# Brefeldin A—A Major Pathogenic Factor of Peanut Pod Rot from *Fusarium neocosmosporiellum*

**DOI:** 10.3390/toxins16120548

**Published:** 2024-12-18

**Authors:** Huiling Wang, Xiaohan Wang, Huiling Han, Quanlin Yu, Xinmiao Tan, Junlong Liu, Yiting Zhao, Weiming Sun

**Affiliations:** 1College of Agriculture and Biotechnology, Hebei Normal University of Science and Technology, Qinhuangdao 066004, China; wanghuiling95@163.com (H.W.); quanlin_yu@aliyun.com (Q.Y.); tanxinmiao2023@163.com (X.T.); liujunlong202412@163.com (J.L.); yitingzhao2023@126.com (Y.Z.); 2Hebei Key Laboratory of Crop Stress Biology, Hebei Normal University of Science and Technology, Qinhuangdao 066004, China; hlhan@hevttc.edu.cn; 3College of Marine Resources and Environment, Hebei Normal University of Science and Technology, Qinhuangdao 066004, China

**Keywords:** peanut pod rot, brefeldin A, *Fusarium neocosmosporiellum*, mycotoxins, phytotoxicity

## Abstract

*Fusarium neocosmosporiellum* is the main pathogen of peanut pod rot in China. To investigate the type of *F. neocosmosporiellum* toxin and its pathogenic mechanism, a macrolide, brefeldin A, was isolated. The structure of the compound was identified by 1D and 2D nuclear magnetic resonance (NMR) and high-resolution electrospray ionization–mass spectrometry (HR-ESI-MS). At the same time, the content of the compound in healthy and diseased peanut capsules was detected, and its plant toxicity to radish, mung bean, rice, and peanut seed radicle elongation and pathogenicity to peanut pod rot were evaluated. The results showed that brefeldin A at 50 μg/mL could significantly inhibit the radicle elongation of rice seeds. Brefeldin A was detected only in pods with peanut rot. Injecting 2 mg/mL brefeldin A solution into peanut pods caused the severe decay of peanut pods at the R3R4 stage, which is consistent with the symptoms of peanut rot.

## 1. Introduction

Peanut (*Arachis hypogaea* L.) is a significant crop representative of oil and cash crops, and its planting area and yield are among the top in the world. However, with the expansion of planting scale and the change in climate environment, peanut diseases have become increasingly prominent, especially peanut pod rot, which poses a serious threat to the development of the peanut industry [1,2].

Peanut pod rot is a widespread soil-borne disease. Once it occurs, it will not only lead to peanut yield reductions but also seriously affect the quality and economic benefits of peanuts [3]. In severe cases, the incidence rate can reach more than 60%, which brings huge economic losses to China’s peanut industry. There are many types of pathogenic fungi of peanut pod rot. These include *Fusarium neocosmosporiellum* (formerly *Neocosmospora vasinfecta*), *Berkeleyomyces rouxiae*, *Pythium* spp., *P. delicense, P. myriotylum*, *Rhizoctonia solani*, *Sclerotinia minor*, *S. sclerotiorum*, *S. rolfssii*, *F. oxysporum*, etc. [3,4,5,6,7,8,9,10]. These pathogens cause varying degrees of damage to peanut plants through different infection routes and pathogenic mechanisms, leading to the decline of fruit yield and quality. The majority of peanut pod rot cases in China are attributed to *F. neocosmosporiellum*. Our research group previously reported that *F. neocosmosporiellum* was the pathogen of peanut pod rot in northern China, and *F. neocosmosporiellum* was isolated and identified as the main pathogen of peanut pod rot in the main peanut-producing areas of Hebei Province [3,11,12]. However, the pathogenesis of peanut pod decay caused by this pathogen has not been reported.

As an important plant pathogen, *Fusarium* sp. can cause root rot, stem rot, ear rot, and so on [12,13]. The pathogenic mechanism of *Fusarium* sp. is complex. By dissolving the defense system of the plant and releasing toxins, it causes symptoms such as wilt and decay of the plant and eventually induces the death of the crop [14]. For example, cell wall degrading enzymes and the pectinases secreted by *Fusarium* sp. can attack the vascular bundle system of plants, decompose plant cell walls, and block ducts, thus interfering with the normal physiological functions of host plants [15]. Fusarium acid harms plants by inhibiting respiration, altering cell permeability, causing electrolyte leakage, interfering with ion balance, and reducing polyphenol oxidase activity [16]. In addition, *Fusarium* sp. can also produce a variety of fungal toxins such as trichothecenes, zearalenone, fumonisins, enniatins, monilifoemin, and beauvericin, contaminating grains and producing adverse effects pertaining to the growth and development of plants and animals [17].

BFA has been isolated from various soils or, more recently, marine fungi and has shown versatile beneficial activities [18]. The compounds were tested for antiproliferative activity in the National Cancer Institute’s 60 cancer cell line screen [19]. The brefeldin A ester derivatives are potential anticancer agents [20]. Brefeldin A (BFA), a natural Arf1 inhibitor, qualifies for extremely superior antitumor activity against HCC [18]. BFA showed significant inhibition on TNFα-induced necroptosis by disrupting necrosome formation and suppressing the phosphorylation of RIPK3 and MLKL [21].

In this study, *F. neocosmosporiellum*, a pathogen of peanut rot disease in Hebei Province, was screened for compound toxins with pathogenic effects through the activity tracking separation method. Then, it was selected as the research object where a toxic metabolite was isolated from it, identified as brefeldin A; the plant’s brefeldin A toxic activity and the pathogenicity of peanut pod rot were then evaluated. This paper describes the isolation, purification, and biological activity of *F. neocosmosporiellum* toxin. *F. neocosmopoliellum’s* pathogenic factors are unclear, and the chemical composition of its metabolites is not yet clear. The toxin produced by *F. neocosmopoliellum* on peanut pod rot is also unclear. We extracted *F. neocosmosporiellum,* the secondary metabolite of which was used to obtain the pathogenic toxin BFA, which was the first to be derived from *F. neocosporoiellum.* The findings of this study offer ideas for further elucidation of the pathogenic mechanism of *F. neocosmosporiellum* causing peanut pod rot and disease prevention and control.

## 2. Results

### 2.1. Structure Elucidation of Brefeldin A *(**1**)*

Compound **1** was obtained in the form of a white, amorphous powder. High-resolution electrospray of the HR-ESI-MS analysis revealed a prominent quasi-molecular ion peak at m/z 279.160 [M-H]^−^ (Appendix A), indicating it was the molecular formula C_16_H_24_O_4_(Calcd. for C_16_H_23_O_4_, 279.164), possessing five degrees of unsaturation. ^1^H-NMR (CD_3_OD, 600 MHz) and ^13^C-NMR (CD_3_OD, 125 MHz) data can be found in Table 1. The ^13^C-NMR (Table 1) of **1** showed one carbonyl (*δ*_C_ 166.95), four sp^2^-hybridized methines (*δ*_C_ 153.70, 136.71, 129.99, 116.36), three oxygenated methines (*δ*_C_ 75.21, 71.81, 71.59), two sp^3^-hybridized methines (*δ*_C_ 51.77, 44.04), five sp^3^-hybridized methylenes (*δ*_C_ 42.08, 40.45, 33.58, 31.57, 26.62), and one methyl (*δ*_C_ 19.69). The ^1^H-NMR spectrum (Table 1) of **1** exhibited similarities to that of brefeldin A (Figure 1). Comprehensive analyses of ^1^H-NMR (Figure 1), ^13^C-NMR (Figure 2), DEPT 135°, ^1^H-^1^H COSY and HMBC spectrum (Appendix A), were consistent with the previous report [22]. Compound 1 (Figure 3) was determined to have the same structure as brefeldin A.

### 2.2. Distribution of Brefeldin A in Matured Peanut Pod

The contents of brefeldin A in different parts of diseased and healthy peanuts are shown in Table 2. Brefeldin A was detected in peanut pods and peanut shells, which were infected by *F. neocosmosporiellum*. The contents of the rot in peanut pods and shells were 0.74 and 0.85 mg/g, respectively. Very little or no brefeldin A was detected in the part of the diseased peanut nutlet. There was no brefeldin A detected in all parts of the healthy peanut either. Therefore, it was speculated that brefeldin A may be one of the causes of peanut pod decay.

### 2.3. Phytotoxicity Assay of Brefeldin A

Brefeldin A was evaluated for its phytotoxic activities on leguminous plants (mung bean and peanut) and non-legume species. Brefeldin A exhibited potent suppressive effects on the elongation of rice seed radicles (Table 3). At 12.5 μg/mL levels, the inhibition ratio was 65.08%. When the concentration rose above 50 μg/mL, the inhibition rate reached above 95%. Brefeldin A exhibited moderate activity against mung bean and radish radicle elongation at a concentration of 200 and 500 μg/mL; the inhibition ratios were at least 70%, respectively. Brefeldin A showed weak peanut seed radicle elongation inhibition activity with a 41.35% inhibition ratio at 1000 μg/mL, and it has poor repeatability. This result may be due to the fact that peanut seeds are larger than other seeds. Due to the limited quality of isolated brefeldin A, no higher concentration was performed on peanut seeds. The photo results of the mung bean seeds treated with brefeldin A can be seen in Figure 4. The photo resultss of the rice bean seeds treated with brefeldin A the results can be seen in Figure 5. The results suggest that brefeldin A has a broad spectrum of inhibitory activity on plant seed radicle growth.

### 2.4. Toxic Activity Assay of Brefeldin A on Peanut Pod

To investigate the effect of brefeldin A on peanut pod decay, two varieties of peanut pod were needled. Yuhanghua 7 is a disease-resistant variety, and Zhonghua 12 is a susceptible variety [23]. Table 4 shows the peanut pod lesion area of different varieties and periods.

With the increasing concentration of brefeldin A, the decay degree of the peanut pod increased in both peanut varieties. However, with the increasing maturity of the peanut pod, the degree of peanut decay decreased (Figure 6). For peanut pods in R3R4 periods, the control group (CK) showed a slight yellow spot at the needled point. At the concentration of 2 mg/mL, the peanut kernel and shell were badly rotted, and the inner part hollowed and turned black in both varieties. At the R5R6 periods, the kernel and shell turned black only, while at the R7R8 periods, the puncture wound turned black only. There was no rot or hollow activity found in the R7R8 periods. The results showed that brefeldin A could significantly cause pod decay when the kernel was not mature. The mature peanut pod was less affected by brefeldin and had higher resistance, which was consistent with the infection period of *F. neocosmosporiellum* on peanuts. In conclusion, the toxin brefeldin A, produced by *F. neocosmosporiellum*, is one of the important causes of peanut pod rot.

At the same concentration of brefeldin A, the decay of Zhonghua 12 is more serious than that of Yuhanghua 7. It was consistent with the disease resistance of the two peanut varieties.

## 3. Discussion and Conclusions

In conclusion, a main mycotoxin, brefeldin A, was identified from the *Fusarium neocosmosporiellum* ferment. The structure of brefeldin A was revealed through NMR and HR-ESI-MS analyses. In the meantime, the distribution of brefeldin A in matured peanut pods was detected, as well as its toxic activity on peanut pods and phytotoxicity on radish, rice, mung bean, and peanut, which were evaluated.

Brefeldin A is a 16-component macrolide compound. This compound was first discovered in *Penicillium decumbens* and showed toxic activity in rats and goldfish and inhibitory activity in wheat germination [24]. In subsequent reports, the compound was isolated from *Penicillium* fungi several times and exhibits a variety of biological activities, such as antifungal, antiviral, antitumor, and acetylcholinesterase inhibitory activities [25,26,27,28]. In addition, brefeldin A was also found in *Alternaria carthami* and *Cylindrocarpon obtusisporum* [29,30]. In this study, brefeldin A was isolated from *F. neocosmosporiellum*, which is the first time that this compound has been found and isolated from *Fusarium* fungi.

The mechanism of action of brefeldin A is not yet clear. BFA inhibits clathrin-dependent endocytosis and ion transport in Chara internodal cells [31]. Straus AJ shows that the ER stress inducer brefeldin A (BFA) causes increased glycosylation of CerS6 and that treatment with BFA causes increased endogenous ceramide accumulation [32]. The pathogenic mechanism of plant diseases has not been reported yet. In practical applications, we will consider blocking the metabolic pathway of BFA to achieve the goal of reducing or not secreting BFA, as well as adjusting the soil environment to change the soil microecology to reduce the number of new *Fusarium neocosmosporiellum* or using antagonistic bacteria [33] to reduce the number of *Fusarium neocosmosporiellum*, thereby reducing the production of this compound.

Peanut is an important oil crop in China. Peanut pod rot seriously harms the yield and quality of peanuts and causes huge economic losses. *F. neocosmosporiellum* (formerly named *Neocosmospora vasinfecta*) was isolated and identified as the dominant pathogen of peanut pod rot in Hebei Province. A variety of toxins, such as neovasinin, neovasinone, neovasinfuranones, neovasipyridones, and vasinfectins, were isolated from *N. vasinfecta*. Both neovasinin and neovasinone could inhibit the elongation of lettuce seed radicles, and vasinfectins A and B could lead to leaf greening of soybean seedlings [34,35,36,37]. All these compounds showed phytotoxic activity, but the cause of peanut pod decay has not been investigated. In this research, HPLC detection was performed on the BFA content in both healthy pods and diseased fruits, while we only detected BFA in diseased fruits. Inoculation of isolated brefeldin A into peanut pods could induce symptoms consistent with peanut pod rot. The above results showed that brefeldin A was one of the main causes of peanut pod rot.

Peanut rot occurs after the formation of peanut pods and mainly harms peanut pods. Most of this disease starts in mid-July when peanut pods are mostly at the R3R4 stage in northern China. At this time, the peanut kernel was not expanded, and spongy tissue was abundant. When brefeldin A was injected into peanut capsules at the R3R4 stage, dark brown spots were formed on the inoculated peanut skins and gradually expanded until the pods and kernels were decomposed and cavities appeared, leaving only fibrous tissue on the pod epiderma, which was the same as the symptoms of early peanut rot. The late-onset diseased peanut pods formed dark brown spots and gradually expanded; the shells did not rot, the interior turned black, and the kernels were intact, which was consistent with the symptoms of the pods during the R7R8 period of brefeldin A injection [3,38]. In conclusion, brefeldin A is one of the main factors of peanut pod decay caused by *Fusarium neocosmosporiellum*, which is of great significance for the future targeted prevention and control of peanut pod rot.

## 4. Materials and Methods

### 4.1. General Experimental Procedures

High-resolution electrospray ionization mass spectrometry (HR-ESI-MS) was recorded on a Thermo Scientific LTQ-Orbitrap XL instrument. ^1^H-, ^13^C-, and 2D nuclear magnetic resonances (NMR) were measured using a Bruker Magnetic Resonance spectrometer (600 M). The high-performance liquid chromatography (HPLC) analysis was performed using a LabAlliance PC2001 instrument with a Model 205 ultraviolet detector (LabAlliance, Tianjin, China) and an analytical reversed-phase Acclaim^TM^ 120 C18 column (4.6 mm × 250 mm, 5 μm; Thermo Scientific, CA, USA). Precoated silica gel GF-254 plates (Qingdao Marine Chemical Co., Ltd., Tsingtao, SD, China) were used for analytical thin-layer chromatography (TLC). Spots were visualized under UV light at 254 nm and 365 nm and sprayed with a 5% sulfuric acid ethanol chromogenic agent.

The silica gel fillers used for separation were purchased from Qingdao Marine Chemical Co., Ltd. The Sephadex G-15 was purchased from GE Healthcare Bio-Sciences AB. The methanol used for the HPLC was chromatography grade, purchased from Thermo Fisher Scientific. Ultrapure water was used throughout the experiment. All other reagents were of analytical grade.

### 4.2. Fungal Materials

The pathogen fungus was isolated from the rot of the peanut pod in our previous study [3]. The fungus was identified as *Fusarium neocosmosporiellum* by morphological and molecular biological methods [3]. The strain was deposited in the College of Marine Resources and Environment, Hebei Normal University of Science and Technology.

### 4.3. Fermentation, Extraction, and Isolation

The fungus was grown on potato dextrose agar (PDA) plates at 28 °C for 5~7 days. Then, four to five agar medium disks (0.5 cm × 0.5 cm) with mycelium were inoculated into 250 mL Erlenmeyer flasks, each containing 100 mL of potato dextrose broth (PDB) medium to prepare the inoculum, incubated on a rotary shaker at 180 rpm and 28 °C for 3 days. The enlarged cultivation was carried out in 30 glass bottles (500 mL), each containing 300 g of corn kernels and 300 mL of ultrapure water. The corn kernel culture medium was inoculated using the inoculum. The fermentation was kept at 25 °C for 60 days before harvest.

The fermented corn kernels substrate was combined and ground, extracted by infusion with MeOH at room temperature three times (10 L for each time). After filtration, the filtrates were combined and evaporated under vacuum at 40 °C by a rotary evaporator to give a dark brown extract residue. The extracted residue was suspended in water and sequentially partitioned with petroleum ether, EtOAc, and *n*-BuOH to obtain their corresponding fractions. According to the preliminary evaluation of phytotoxicity, the EtOAc fraction showed inhibitory activity against mung bean seed radicle elongation. Then, the EtOAc extracts were combined and concentrated under vacuum at 40 °C to obtain a brown residue (7.21 g).

The EtOAc extract was subjected to column chromatography (CC) over silica gel (100–200 mesh) eluting with a gradient of CH_2_Cl_2_-MeOH (100:0~0:100). TLC was used to detect each fraction, and similar ones were combined to obtain eighteen fractions (Frs. 1~18). Fr. 9 (2.18 g) was further subjected to silica gel CC eluting with a gradient of CH_2_Cl_2_-MeOH (0:100~100:0) to yield 12 subfractions (Fr. 9-1~Fr. 9-12). Fr. 9-4 was further purified by preparative thin-layer chromatography (PTLC) using CH_2_Cl_2_-MeOH (95:5) and recrystallization to afford compound **1** (88 mg). The structures of compound **1** were identified using the NMR and HR-ESI-MS.

### 4.4. HPLC Analysis of Brefeldin A

The samples analyzed contained rotten peanut pods, shells, kernels, and healthy pods. The samples were ground into powder. Furthermore, 10 g samples were soaked in MeOH and extracted in an ultrasonic bath at room temperature three times (3 × 50 mL, 30 min for each time). The extracted solution was combined and concentrated under vacuum by a rotary evaporator to dryness at 40 °C. The residue was dissolved in MeOH and fixed to 1 mg/mL. The sample solution was filtered through a microporous filter (pore size, 0.22 μm, Tianjin Jinteng Experimental Equipment Co., Ltd. Tianjin, China,) before analysis.

For preparing the brefeldin A standard solution, 1 mg of purified brefeldin A was dissolved in 1 mL of MeOH to obtain the stock solution of 1 mg/mL. The stock was further diluted with MeOH to obtain a series of concentrations (500.0, 250.0, 125.0, and 62.5 μg/mL). All the solutions were filtered through a microporous filter (pore size of 0.22 μm) before analysis.

The filtered solution was analyzed by a LabAlliance PC2001 high-performance liquid chromatograph system, with a flow rate of 1 mL/min, eluted with 70% MeOH (*v*/*v*) and 30% H_2_O for 30 min. The detection of the UV wavelength was at 254 nm. The sample injection volume was 10 μL.

The method validation used for the HPLC quantitative analysis of brefeldin A has been reported [39]. The HPLC conditions in this study proceeded with minor modifications. The calibration curve of brefeldin A was Y = 1546.3X − 12036, R² = 0.9998, where Y is the peak area, and X is the injection volume (μL) of the analyte. The results showed good linearity in the range of 62.5–1000 μg/mL in the sample injected. The HPLC profiles of standard brefeldin A and the sample extract are in Figure 7, respectively. Brefeldin A in the sample extract was identified by comparison of the retention time.

### 4.5. Inhibitory Activity Assay on Radicle Elongation of Plant Seeds

The phytotoxicity of brefeldin A was evaluated by the inhibitory activity of plant seeds’ radicle elongation. Radish (*Scrophularia ningpoensis* Hemsl), rice (*Oryza sativa* L), mung bean (*Vigna radiata*), and peanut (*Arachis hypogaea* L) seeds were used for the assay. The radish seeds came from Yinong Seed Industry Co., Ltd. in Fuyu City, China. The rice seeds came from Hunan Jinjin Nongfeng Seed Industry Co., Ltd. (Hunan, China). The mung bean seeds came from Hou Ma Rui De Feng Seed Industry Co., Ltd. (Shanxi, China). Peanut seed laboratory retention.

The inhibitory activity assay on radicle elongation was performed as described previously [40] with some modifications. For surface disinfection, the seeds were soaked in 75% EtOH for 30 s and rinsed with sterile water; then, the seeds were transferred into 5% sodium hypochlorite for 3 min and rinsed with sterile water until there was no sodium hypochlorite residue. The plant seeds were sown in a Petri dish moisturized at 25 °C until germination. A total of 100 μL of the working solution was added to the germinated seeds and incubated in a moist chamber at 25 °C in the dark. A total of 2 mg of purified brefeldin A was dissolved in 1 mL of 0.5% MeOH-H_2_O solution to obtain the stock solution (2 mg/mL). Working solutions were obtained by further diluting the stock solution to a series of concentrations (1500, 1000, 500, 200, 100, 50, 25, and 12.5 μg/mL) with the 0.5% MeOH-H_2_O solution. The 0.5% MeOH-H_2_O solution was used as the negative control. The length of each radicle was measured after treatment for 48 h. The inhibition of radicle elongation was calculated as follows: Inhibition rate (%) = [(*L_c_* − *L_t_)/L_c_*] × 100%, where *L_c_* is the radicle length of the control group, and *L_t_* is the length of the treated group.

All experiments were performed with three replicates, and the results were represented by their mean values and standard deviations. The data were carried out using analysis of variance (one-way ANOVA) to detect significant differences by ANOVA using SPSS version 26. Different letters indicated that the data were significantly different at *p* ≤ 0.05.

### 4.6. Toxic Activity Assay of Brefeldin A on Peanut Pods at Different Stages

Based on the visually observable reproductive (R) events, the peanut growth stage can be divided into nine stages from R1 to R9, as reported [41]. The R3 stage is the beginning pod, and R8 is harvest maturity. Therefore, stage R3 to R8 peanut pods were used for the assay. Each stage does not have a clear dividing point; they were classified into three stages (R3R4, R5R6, and R7R8) so as to use statistics easily. The R3R4 stage is the beginning pod to full pod, the R5R6 stage is the beginning seed to full seed, and R7R8 is the beginning of maturity to harvest maturity.

The peanut pods were separated from plants at different stages. The pods were punctured with sterile needles containing 100 μL of the working solution. A total of 2 mg of purified brefeldin A was dissolved in 1 mL of 0.5% MeOH-H_2_O solution and then diluted to obtain a working solution (2, 1, and 0.5 mg/mL). The 0.5% MeOH-H_2_O solution was used as the negative control. Symptom appearance was observed daily up to 3 days after the acupuncture experiments. All experiments were performed with three replicates. The lesion area was measured by PS version 2023. The data were carried out using analysis of variance (one-way ANOVA) to detect significant differences by ANOVA using SPSS version 26.

## Figures and Tables

**Figure 1 toxins-16-00548-f001:**
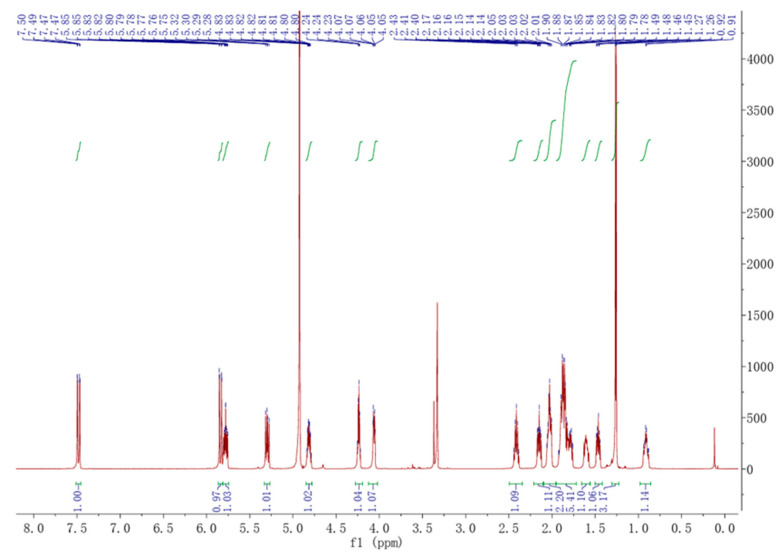
The ^1^H NMR of brefeldin A.

**Figure 2 toxins-16-00548-f002:**
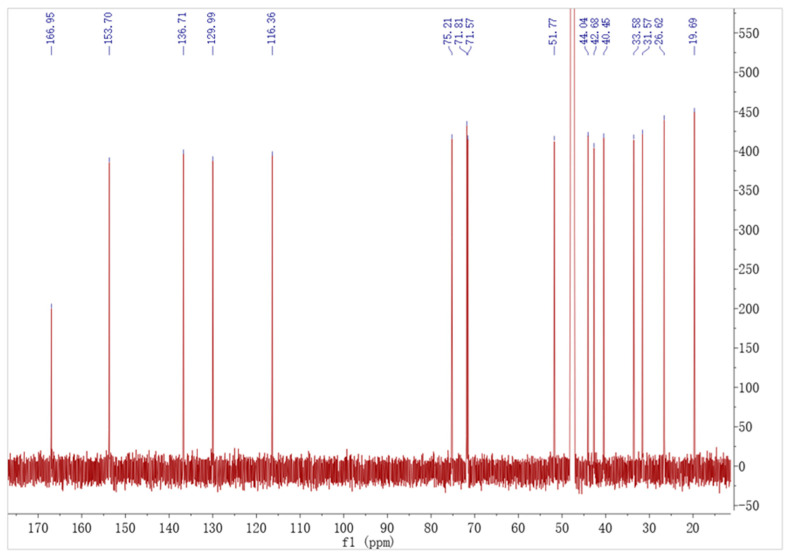
The ^13^C NMR of brefeldin A.

**Figure 3 toxins-16-00548-f003:**
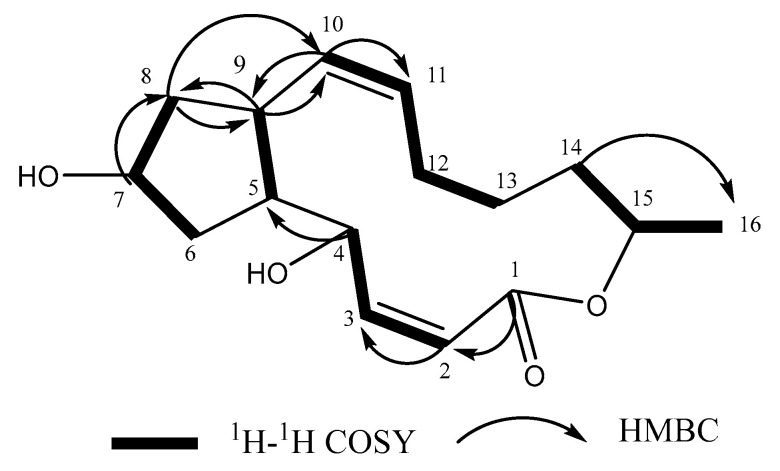
Structure of brefeldin A (**1**) and its correlation with ^1^H-^1^H COSY and HMBC.

**Figure 4 toxins-16-00548-f004:**
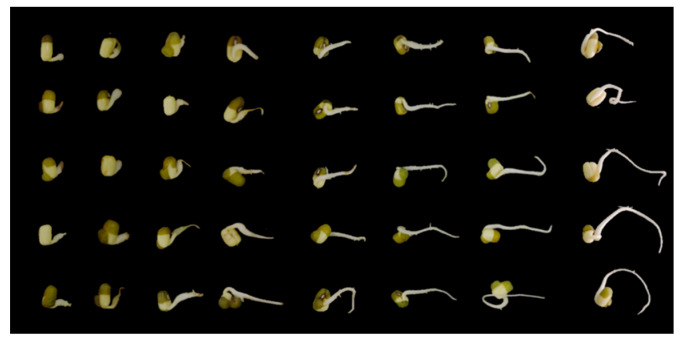
Inhibitory activity of brefeldin A (**1**) on the radicle elongation of the mung bean seed. The mung bean was treated at concentrations of 1000, 500, 200, 100, 50, 25, 12.5, and 0 μg/mL, from left to right.

**Figure 5 toxins-16-00548-f005:**
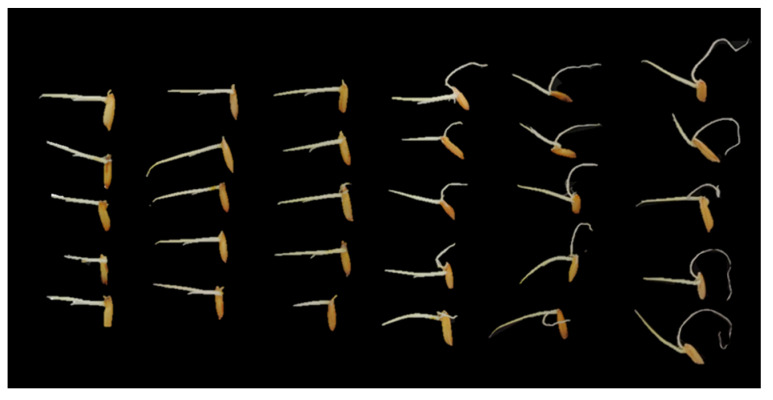
Inhibitory activity of brefeldin A (**1**) on the radicle elongation of the rice bean seed. The rice bean was treated at concentrations of 200, 100, 50, 12.5, 6.25, and 0 μg/mL, from left to right.

**Figure 6 toxins-16-00548-f006:**
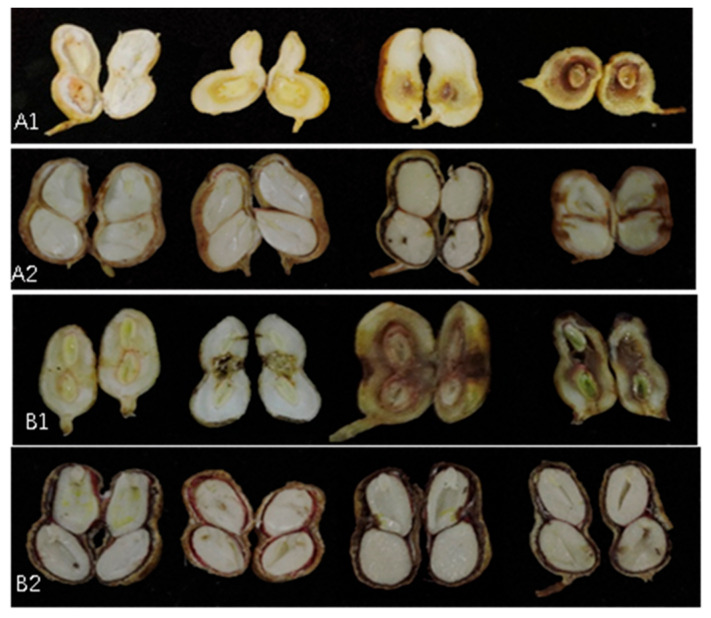
Toxicity of brefeldin A against different periods for peanut pods. (**A1**) is the photo of Yuhanghua 7 in R3R4 periods for peanut pods. (**A2**) is the photo of Yuhanghua 7 in R7R8 periods for peanut pods. (**B1**) is the photo of Zhonghua 12 in R3R4 periods for peanut pods. (**B2**) is the photo of Zhonghua 12 in R7R8 periods for peanut pods. All the peanut pods were treated with brefeldin A at concentrations of 0, 0.5, 1, and 2 mg/mL, from left to right.

**Figure 7 toxins-16-00548-f007:**
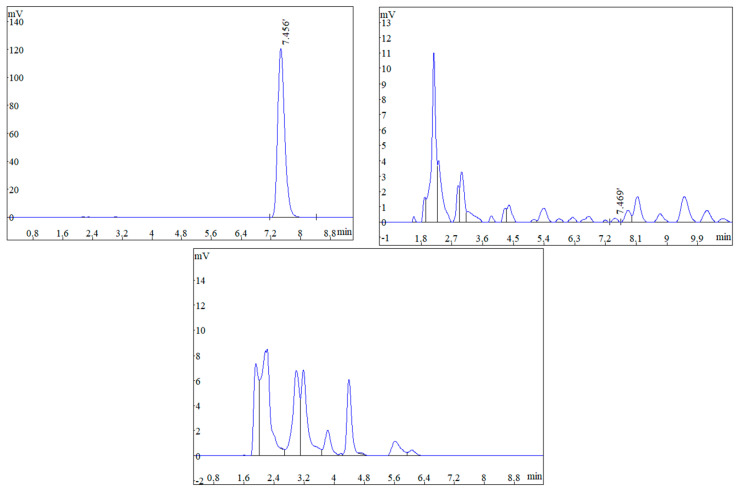
HPLC profile of brefeldin A (1000 μg/mL), HPLC profile of rotten pods, and HPLC profile of healthy pods.

**Table 1 toxins-16-00548-t001:** ^1^H-NMR and ^13^C-NMR statistics on brefeldin A (**1**) in CD_3_OD (*δ* in ppm, *J* in Hz).

Position	Measured Data		Reference Data [18]	
	*δ*_C_ ^a^	*δ*_H_, mult. (*J* in Hz) ^b^	*δ*_C_ ^c^	*δ*_H_, mult. (J in Hz) ^d^
C(1)	166.95	-	168.38	-
CH(2)	116.36	5.84, dd (15.6, 1.8)	117.78	5.85, dd (15.6,1.9)
CH(3)	153.70	7.48, dd (15.6,3.0)	155.12	7.49, dd (15.6,3.0)
CH(4)	75.21	4.08–4.04, m	76.63	4.07–4.01, m
CH(5)	51.77	1.92–1.76, m	53.18	1.92–1.72, m
CH_2_(6)	40.45	2.06–2.01, m	41.85	2.06–1.96, m
		1.92–1.76, m		1.92–1.72, m
CH(7)	71.81	4.26–4.19, m	73.21	4.25–4.18, m
CH_2_(8)	42.08	2.15, ddd (13.8, 9.0, 5.4)	44.09	2.12, ddd (13.6, 8.6, 5.4)
		1.49–1.44, m		1.48–1.40, m
CH(9)	44.04	2.41, q(8.4)	45.46	2.39, q (8.7)
CH(10)	136.71	5.3, dd (15.0,9.6)	138.13	5.2, dd (15.0,9.6)
CH(11)	129.99	5.78, ddd (15.0, 10.2, 4.8)	131.41	5.79, ddd (15.2, 10.2, 4.7)
CH_2_(12)	31.57	2.06–1.96, m	32.97	2.06–1.96, m
		1.92–1.76, m		1.92–1.72, m
CH_2_(13)	26.62	1.92–1.76, m	28.01	1.92–1.72, m
		0.94–0.87, m		1.00–0.85, m
CH_2_(14)	33.58	1.92–1.76, m	34.99	1.92–1.72, m
		1.63–1.57, m		1.63–1.52, m
CH(15)	71.59	4.84–4.78, m	72.98	4.83–4.75, m
CH_3_(16)	19.69	1.26, d (6)	21.06	1.27, d (6.3)

Note: ^a^ Recorded at 125 MHz. ^b^ Recorded at 600 MHz. ^c^ Recorded at 100 MHz. ^d^ Recorded at 600 MHz. -, not detected.

**Table 2 toxins-16-00548-t002:** Contents of brefeldin A in peanut.

Part of Peanut	Extract Quality (mg)	Brefeldin A Content (mg/g)
Pod rot of peanuts		
peanut pod	783.33 ± 10.22	0.74 ± 0.02
peanut shell	472.37 ± 9.93	0.85 ± 0.09
peanut nutlet	2409.36 ± 24.99	-
Healthy peanuts		
peanut pod	1574.07 ± 36.81	-
peanut shell	941.67 ± 28.12	-
peanut nutlet	1103.57 ± 17.96	-

Note: -, not detected.

**Table 3 toxins-16-00548-t003:** Inhibitory activity of brefeldin A on plant seeds radicle elongation.

Concentration (μg/mL)	Inhibition Ratio (%)			
	Radish	Rice	Mung Bean	Peanut
1500	-	-	-	48.12 ± 21.65 a
1000	72.64 ± 13.30 a	-	-	41.35 ± 22.21 a
500	71.70 ± 8.60 a	-	-	15.04 ± 33.60 a
200	46.23 ± 13.73 b	-	72.26 ± 6.63 a	-
100	16.04 ± 11.27 c	97.89 ± 2.83 a	57.24 ± 16.86 b	-
50	-	95.46 ± 2.56 a	45.76 ± 16.54 b	-
25	-	84.59 ± 7.08 a	28.62 ± 24.30 c	
12.5	-	65.08 ± 16.26 b		

Note: -, not tested. Different letters indicate significant differences among treatments in each column, including different concentrations at *p* ≤ 0.05.

**Table 4 toxins-16-00548-t004:** Toxicity of brefeldin A against different periods for peanut pods.

Peanut Variety	Concentration (mg/mL)	Lesion Area (cm^2^)		
		R3R4	R5R6	R7R8
Zhonghua 12	0	0.29 ± 0.21 a	0.88 ± 0.39 a	0.35 ± 0.13 a
	0.5	0.28 ± 0.20 a	0.98 ± 0.44 a	2.19 ± 0.73 ab
	1	0.60 ± 0.15 a	1.84 ± 1.13 ab	3.06 ± 1.72 ab
	2	2.12 ± 1.26 b	2.41 ± 1.05 b	1.77 ± 0.90 b
Yuhanghua 7	0	0.07 ± 0.03 a	0.02 ± 0.01 a	0.01 ± 0.00 a
	0.5	0.25 ± 0.18 ab	1.64 ± 0.40 b	2.03 ± 1.28 b
	1	1.29 ± 0.03 b	2.43 ± 1.53 b	2.57 ± 0.89 b
	2	0.86 ± 0.34 c	2.38 ± 1.40 b	3.34 ± 1.67 b

Note: Different letters indicate significant differences among treatments in each column, including different varieties and concentrations at *p* ≤ 0.05.

## Data Availability

The original contributions presented in this study are included in this article and Appendix A. Further inquiries can be directed to the corresponding authors.

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
