# Peer review of "Brefeldin A—A Major Pathogenic Factor of Peanut Pod Rot from Fusarium neocosmosporiellum"

_toxins, 2024, doi:10.3390/toxins16120548_

Round 1

Reviewer 1 Report

Comments and Suggestions for Authors

Thank you so much for your quick action on the prepared manuscript number "Toxins-3304429" "Brefeldin A, a Major Pathogenic Factor of Peanut Pod Rot from Fusarium neocosmosporiellum". My first vision, this can be refined after revision:

1-      To make the article more edited and clear, it is suggested that professionals from native speakers of English should revise the language.

2-      The introduction must be efficient, and should have more fresh data about the biological activity of Brefeldin A (cyclic carboxylic esters).

3-      At the end of the introduction section, the objective of the work should be more clearly stated.

4-      Different seeds were chosen for germination test, but toxicity only done on Peanut, Why?

5-      Spectroscopy figures (1HNMR and 13CNMR) must be included in the text

6-      EtoAc fraction was chosen for phytotoxic according to its activity against mung bean seeds radicle elongation, however its moderate activity in comparison with rice seeds, why (and need reference to support this chosen)?  

7-      Section 2.3: the authors must be Add a figure of the inhibitory activity of Brefeldin A on the other seeds used in the study

8-      Section 3: The data presented hides the amount of work performed. It is visible that the work is good and has large results, but the discussion was too expressive and lacked the explanation of the phenomenon they observed, the authors should discuss the results more deeply with recent references (2020-2024).

9-      Mod of action of active compound (Brefeldin A) should be more clearly and accurate

10-  Improve tables and charts.

11-  Section 4 (Material and Methods): Figure 4. This chart only for HPLC profile of Standard Brefeldin A, but sample extract, where?  

12-  Where the conclusions section? Should be written to highlight most important ones and conclude main aspects of the work, implications, and future prospects.

13-  In the reference section: References must be updated.

Comments on the Quality of English Language

It is suggested that professionals from native speakers of English should revise the language.

Author Response

Comments 1: To make the article more edited and clear, it is suggested that professionals from native speakers of English should revise the language.

Response 1: Thank you for pointing this out. We ask native English speakers to revise the language.

Comments 2: The introduction must be efficient, and should have more fresh data about the biological activity of Brefeldin A (cyclic carboxylic esters).

Response 2: Agree. We have accordingly revised introduction

Comments 3: At the end of the introduction section, the objective of the work should be more clearly stated.

Response 3: Agree. We have accordingly revised introduction

Comments 4: Different seeds were chosen for germination test, but toxicity only done on Peanut, Why?

Response 4: Because this study is dedicated to investigating the pathogenic toxin of the peanut pot rot pathogen F. neocosmosporiellum, BFA was extracted from the pathogenic fungus to investigate whether it is a pathogenic factor.

Comments 5: Spectroscopy figures (1HNMR and 13CNMR) must be included in the text

Response 5: Thank you for pointing this out. I to be added to the text

Comments 6: EtoAc fraction was chosen for phytotoxic according to its activity against mung bean seeds radicle elongation, however its moderate activity in comparison with rice seeds, why (and need reference to support this chosen)? 

Response 6: We used the activity tracking method for the experiment and ultimately chose the active EtoAc. Both mung beans and rice have undergone experiments on the elongation activity of embryonic roots. The reason for choosing to display photos of mung beans is that mung beans and peanuts belong to the same legume family and have a closer genetic relationship. Moreover, as the concentration of BFA increases, the inhibition of embryonic root elongation gradually strengthens, and the experimental results have significant correlation. It has a significant inhibitory effect on the elongation of rice seed embryonic roots, and the inhibitory effect is obvious at low concentrations, making it difficult to show a trend.

Comments 7: Section 2.3: the authors must be Add a figure of the inhibitory activity of Brefeldin A on the other seeds used in the study

Response 7: Thank you for pointing this out. I have added experimental pictures of rice.

Comments 8: Section 3: The data presented hides the amount of work performed. It is visible that the work is good and has large results, but the discussion was too expressive and lacked the explanation of the phenomenon they observed, the authors should discuss the results more deeply with recent references (2020-2024).

Response 8: Thank you for pointing this out. I have added relevant content.

Comments 9: Mod of action of active compound (Brefeldin A) should be more clearly and accurate

Response 9: Thank you for pointing this out. I have added relevant content.

Comments 10: Improve tables and charts.

Response 10: Thank you for pointing this out. We have made modifications.

Comments 11: Section 4 (Material and Methods): Figure 4. This chart only for HPLC profile of Standard Brefeldin A, but sample extract, where? 

Response 11: Thank you for pointing this out. We added HPLC spectra of diseased and healthy fruits to demonstrate the presence of the same compound.

Comments 12: Where the conclusions section? Should be written to highlight most important ones and conclude main aspects of the work, implications, and future prospects.

Response 12: The content of Section 3 is conclusion and discussion, and we have made modifications

Comments 13: In the reference section: References must be updated.

Response 13: We have updated the references.

Reviewer 2 Report

Comments and Suggestions for Authors

This manuscript investigates the isolation and pathogenic role of brefeldin A from Fusarium neocosmosporiellum as a contributor to peanut pod rot. Although the study addresses a topic of agricultural significance, several critical issues regarding experimental design, clarity of objectives, quality of English, and data interpretation weaken its suitability for publication in its current form.

The study lacks a robust rationale or literature background, especially in the Introduction. The manuscript states that brefeldin A was isolated but does not clarify its novelty in the context of peanut pathogens, nor does it sufficiently explore previous findings on its pathogenicity in similar studies. This lapse limits the contribution of the manuscript to existing knowledge.

The research aims are not clearly described. The manuscript should benefit from clearly stating its objectives, such as whether it is focused on identifying the compound’s role in pathogenicity, understanding its mechanism, or assessing its potential as a biomarker for infection.

The pathogenicity tests lack comprehensive controls and do not thoroughly validate that brefeldin A is solely responsible for peanut pod rot symptoms. Comparisons with untreated controls, other pathogens, and toxin standards would help clarify its specific role.

The phytotoxicity assays use a limited number of plant species, with no apparent justification for the choice of test species. Including more relevant agricultural crops or field conditions would strengthen the findings.

Although brefeldin A is quantified, there are limited methodological details about the calibration and validation of its detection. The quantification methods, particularly for HPLC, require more rigorous validation and reporting to ensure reproducibility.

The data on concentration of brefeldin A and pathogenicity are presented ambiguously, making it difficult to interpret its exact role in disease development. Including a quantitative assessment of disease severity relative to brefeldin A concentration could provide valuable understanding.

English of the manuscripts is substandard, containing multiple typographical, grammatical, and syntactical errors. A thorough language revision by a proficient editor is essential to improve sentence structure, coherence, and flow.

In its current form, the manuscript is not suitable for publication due to significant flaws in methodology, unclear research objectives, and inadequate English language quality. Addressing these issues would markedly improve the scientific rigor, readability, and relevance of the study, making it more suitable for publication.

Comments on the Quality of English Language

English of the manuscripts is substandard, containing multiple typographical, grammatical, and syntactical errors. A thorough language revision by a proficient editor is essential to improve sentence structure, coherence, and flow.

Author Response

Comments 1: This manuscript investigates the isolation and pathogenic role of brefeldin A from Fusarium neocosmosporiellum as a contributor to peanut pod rot. Although the study addresses a topic of agricultural significance, several critical issues regarding experimental design, clarity of objectives, quality of English, and data interpretation weaken its suitability for publication in its current form.

Response 1: Thank you for pointing this out. We have accordingly revised .

Comments 2: The study lacks a robust rationale or literature background, especially in the Introduction. The manuscript states that brefeldin A was isolated but does not clarify its novelty in the context of peanut pathogens, nor does it sufficiently explore previous findings on its pathogenicity in similar studies. This lapse limits the contribution of the manuscript to existing knowledge.

Response 2: BFA has been isolated for the first time from the metabolic products of Fusarium neocosmosporiellum, which causes peanut pot rot disease. This discovery is novel. There have been many studies on the pathogenic bacteria of peanut pot rot disease, but no one has isolated its metabolic products and studied the pathogenic factors of this fungus causing peanut pot rot disease from a toxin perspective.

Comments 3: The research aims are not clearly described. The manuscript should benefit from clearly stating its objectives, such as whether it is focused on identifying the compound’s role in pathogenicity, understanding its mechanism, or assessing its potential as a biomarker for infection.

Response 3: The article clearly states that the research focuses on finding pathogenic factors related to toxins in Fusarium neocosmosporiellum and isolating the compound BFA with pathogenic effects.

Comments 4: The pathogenicity tests lack comprehensive controls and do not thoroughly validate that brefeldin A is solely responsible for peanut pod rot symptoms. Comparisons with untreated controls, other pathogens, and toxin standards would help clarify its specific role.

Response 4: BFA is not the only cause of symptoms of peanut pod rot disease. Fusarium neocosmosporiellum is one of the pathogenic bacteria of peanut pod rot disease, and its pathogenic mechanism is complex, not caused by a single cause. The needle puncture test in the article caused symptoms of peanut pod rot disease because the intact peanut pods did not show symptoms. BFA works inside the pods, and the cell wall degrading enzyme secreted by Fusarium destroys the fruit shell, causing BFA to enter the pods and cause disease. These contents are mentioned in the text.

Comments 5: The phytotoxicity assays use a limited number of plant species, with no apparent justification for the choice of test species. Including more relevant agricultural crops or field conditions would strengthen the findings.

Response 5: Our experimental species selection is diverse, including leguminous plants, non leguminous plants, monocotyledonous plants, etc. We screened four plants with obvious trends based on the strength of inhibition rate and whether there is a significant correlation with concentration: peanuts, which are the pathogenic bacteria of peanut fruit rot, mung beans in the legume family, water radish in the cruciferous family, and rice in the monocotyledonous family.

Comments 6: Although brefeldin A is quantified, there are limited methodological details about the calibration and validation of its detection. The quantification methods, particularly for HPLC, require more rigorous validation and reporting to ensure reproducibility.

Response 6: Thank you for pointing this out. We have accordingly revised .

Comments 7: The data on concentration of brefeldin A and pathogenicity are presented ambiguously, making it difficult to interpret its exact role in disease development. Including a quantitative assessment of disease severity relative to brefeldin A concentration could provide valuable understanding.

Response 7: Hello, our experimental results indicate that the higher the concentration of BFA, the more severe the symptoms exhibited by the pods, which precisely demonstrates the pathogenicity of BFA and confirms that BFA does indeed play a role in peanut fruit rot.

Comments 8: English of the manuscripts is substandard, containing multiple typographical, grammatical, and syntactical errors. A thorough language revision by a proficient editor is essential to improve sentence structure, coherence, and flow.

Response 8: Thank you for pointing this out. We ask native English speakers to revise the language.

Comments 9: In its current form, the manuscript is not suitable for publication due to significant flaws in methodology, unclear research objectives, and inadequate English language quality. Addressing these issues would markedly improve the scientific rigor, readability, and relevance of the study, making it more suitable for publication.

Response 9: Thank you for pointing this out. We have made comprehensive revisions to the article to ensure your satisfaction. Thank you for your feedback and suggestions.

Reviewer 3 Report

Comments and Suggestions for Authors

The paper submitted to me for review, entitled "Brefeldin A, a Major Pathogenic Factor of Peanut Pod Rot from Fusarium neocosmosporiellum", is promising and provides new information on the pathogenicity of F. neocosmosporiellum, but requires refinement, particularly in relation to the research hypotheses, interpretation of results and methodology. Clarification of these aspects will increase the scientific value of the study and facilitate its uptake by readers.

Introduction

The problem of pod rot in peanuts caused by Fusarium neocosmosporiellum is mentioned in the introduction, but a clearly defined research hypothesis is missing. Is the aim of the study to prove that brefeldin A is an important pathogenic factor? Is the aim chemical characterisation and toxic activity? Please clarify.

The study assumes that the presence of brefeldin A in rotten peanut pods is evidence of its role in pathogenesis. However, it is not explained whether other mycotoxins could also play an important role.

Materials and methods

It is not clear from the section on quantitative HPLC analysis (4.4) whether a calibration was performed for each sample and how possible interferences from other metabolites in the samples were controlled.

The toxic activity study lacks appropriate controls. I would suggest using e.g. other macrolides or chemical analogues as a comparison.

Conclusions and discussion

Although brefeldin A has been identified as the toxin associated with gangrene, there is no evidence that it is the sole cause of the disease. The results obtained only suggest a correlation.

The conclusions could link the results more clearly to practical applications, e.g. the possible development of crop protection products. It might be worth adding a discussion of the possible application of knowledge about brefeldin A in disease management in the conclusions section.

Author Response

Comments 1: The problem of pod rot in peanuts caused by Fusarium neocosmosporiellum is mentioned in the introduction, but a clearly defined research hypothesis is missing. Is the aim of the study to prove that brefeldin A is an important pathogenic factor? Is the aim chemical characterisation and toxic activity? Please clarify.

Response 1: Thank you for pointing this out. The purpose of this study is to investigate whether toxins are involved in pathogenic fungi and to clarify whether the screened pathogenic toxin compounds are associated with pathogenicity.

Comments 2: The study assumes that the presence of brefeldin A in rotten peanut pods is evidence of its role in pathogenesis. However, it is not explained whether other mycotoxins could also play an important role.

Response 2: Thank you for pointing this out. In this study, BFA was isolated from the secondary metabolites of Fusarium neocosmosporiellum, the pathogenic bacterium of peanut pod rot disease. During the isolation process of the pathogenic fungus, this fungus was found to be the main pathogen in the peanut production area of Hebei Province, and no other pathogenic fungi were isolated. Therefore, no other fungi were studied.

Comments 3: It is not clear from the section on quantitative HPLC analysis (4.4) whether a calibration was performed for each sample and how possible interferences from other metabolites in the samples were controlled.

Response 3: Thank you for pointing this out. We will make improvements to this part to reduce errors.

Comments 4: The toxic activity study lacks appropriate controls. I would suggest using e.g. other macrolides or chemical analogues as a comparison.

Response 4: Thank you for pointing this out. The purpose of this study is to investigate whether BFA are involved in pathogenic.In this study, BFA was isolated from the secondary metabolites of Fusarium neocosmosporiellum, the pathogenic bacterium of peanut pod rot disease. Not for studying the pathogenic effects of macrolide compounds.

Comments 5: Although brefeldin A has been identified as the toxin associated with gangrene, there is no evidence that it is the sole cause of the disease. The results obtained only suggest a correlation.

Response 5: BFA is not the only cause of symptoms of peanut pod rot disease. Fusarium neocosmosporiellum is one of the pathogenic bacteria of peanut pod rot disease, and its pathogenic mechanism is complex, not caused by a single cause. The needle puncture test in the article caused symptoms of peanut pod rot disease because the intact peanut pods did not show symptoms. BFA works inside the pods, and the cell wall degrading enzyme secreted by Fusarium destroys the fruit shell, causing BFA to enter the pods and cause disease. These contents are mentioned in the text.

Comments 6: The conclusions could link the results more clearly to practical applications, e.g. the possible development of crop protection products. It might be worth adding a discussion of the possible application of knowledge about brefeldin A in disease management in the conclusions section.

Response 6: Thank you for pointing this out. We have accordingly revised by conclusions and discussion.

Reviewer 4 Report

Comments and Suggestions for Authors

The present study describes the pathogenic role of brefeldin A, a metabolite of Fusarium neocosmosporiellum, in peanut pod rot disease in China. The authors employed proper analytical methods, including NMR and HR-ESI-MS, to identify and characterize brefeldin A and test the toxicity in various plant seeds. In general, the work is well described but reading the paper i have the impression that the significance of these findings are not highlighted  properly in Discussion (so they don't add much to the current knowledge as brefeldin A was found in numerous other pathogens). 

Suggestions

Please check thoroughly the paper for syntax/grammar errors.

L46. change Fusarium to Fusaric

Fig2. The seedlings are not in line with the concentrations

Paragraph 2.2. This assay on distribution of Brefeldin A in mature pods infected by F. neocosmosporiellum is not described in Methods

Paragraph 2.4. The selected varieties should be described in methods.

Conclusions and Discussion should be strengthen. 

L165. 'In this research, brefeldin A contents were detected in rotten and healthy peanuts,and the presence of brefeldin A was detected only in rotten peanuts.' needs rephrase

Paragraph 4.2. I cannot find the paper [3] to check with the isolate. Since a molecular assay for the detection took place is the sequence deposited in a data base?

L210 and 213. seed culture: it is not clear (maybe inoculum?)

L212 : instead of corn maybe it is best refer as corn kernels?

Comments on the Quality of English Language

The paper should be checked thoroughly for syntax/grammar errors.

eg. L40 'As an important plant pathogen, Fusarium sp. can cause root rot, stem rot, ear rot and so on' the expression so on is not appropriate

Author Response

Comments 1: Please check thoroughly the paper for syntax/grammar errors.

Response 1: Thank you for pointing this out. We ask native English speakers to revise the language.

Comments 2: L46. change Fusarium to Fusaric

Response 2: ”Fusaric "is a compound, and" Fusarium "is the Latin scientific name in the book of Fusarium

Comments 3: Fig2. The seedlings are not in line with the concentrations

Response 3: Thank you for pointing this out. We have made modifications to the caption.

Comments 4: Paragraph 2.2. This assay on distribution of Brefeldin A in mature pods infected by F. neocosmosporiellum is not described in Methods

Response 4: Section 2.2 mainly tested the content of BFA to prove that BFA is a pathogenic factor, present in diseased fruits and not detected in healthy pods. The distribution of BFA in pods has not been studied.

Comments 5: Paragraph 2.4. The selected varieties should be described in methods.

Response 5: Thank you for pointing this out. Yuhanghua 7 is disease-resistant variety, Zhonghua 12 is sus-ceptible variety. Relevant references have been added.

Comments 6: Conclusions and Discussion should be strengthen.

Response 6: Thank you for pointing this out. We have made modifications to this section.

Comments 7: L165. 'In this research, brefeldin A contents were detected in rotten and healthy peanuts,and the presence of brefeldin A was detected only in rotten peanuts.' needs rephrase

Response 7: Thank you for pointing this out. We have made modifications to this section.

Comments 8: Paragraph 4.2. I cannot find the paper [3] to check with the isolate. Since a molecular assay for the detection took place is the sequence deposited in a data base?

Response 8: Thank you for pointing this out. This section has been published in an article, and we have annotated the references in the text.

Comments 9: L210 and 213. seed culture: it is not clear (maybe inoculum?)

Response 9: Thank you for pointing this out. Yes, it's an inoculum. I have made the modifications.

Comments 10: L212 : instead of corn maybe it is best refer as corn kernels?

Response 10: Thank you for pointing this out. We are using corn kernels, thank you for pointing out my expression error.

Round 2

Reviewer 1 Report

Comments and Suggestions for Authors

No comments

Comments on the Quality of English Language

The English could be improved to more clearly express the research.

Author Response

Comments 1: No comments

Response 1: Thank you for considering my manuscript, and I appreciate the absence of further comments as it suggests the content is on the right track. However, I understand the importance of refining the linguistic presentation. Thank you once again for your time and guidance. I look forward to the next steps in the review process.

Response to Comments on the Quality of English Language

Point 1: I will revise the English expression thoroughly to ensure clarity, coherence, and professionalism. This includes checking for grammatical errors, enhancing sentence structures, and ensuring the terminology is precise. I aim to make the manuscript as impactful as possible through improved language.

Reviewer 2 Report

Comments and Suggestions for Authors

A thorough critical review of the revised version of the manuscript indicates that all the concerns and queries raised during the evaluation of the initial submission have been addressed meticulously. The authors have demonstrated diligence in incorporating the suggested revisions, resulting in significant improvements to the overall quality and presentation of the manuscript.

The concerns regarding the substandard English, including typographical, grammatical, and syntactical errors, were addressed. A thorough language revision resulted in significant improvements to the sentence structure, coherence, and overall flow of the manuscript.

The revised version reflects enhanced clarity, coherence, and a more robust discussion of the findings of the study, making it a notable advancement compared to the original submission.

Author Response

Comments 1: A thorough critical review of the revised version of the manuscript indicates that all the concerns and queries raised during the evaluation of the initial submission have been addressed meticulously. The authors have demonstrated diligence in incorporating the suggested revisions, resulting in significant improvements to the overall quality and presentation of the manuscript.

The concerns regarding the substandard English, including typographical, grammatical, and syntactical errors, were addressed. A thorough language revision resulted in significant improvements to the sentence structure, coherence, and overall flow of the manuscript.

The revised version reflects enhanced clarity, coherence, and a more robust discussion of the findings of the study, making it a notable advancement compared to the original submission.

Response 1: Thank you very much for your thorough and critical review of the revised manuscript. We are pleased to know that our efforts in addressing your concerns have been recognized. Your valuable suggestions have significantly improved the quality and clarity of our work. We deeply appreciate your time and guidance.

Reviewer 3 Report

Comments and Suggestions for Authors

Thank you for responding to my comments. In my opinion, the article is suitable for publication in its current form.

Author Response

Comments 1: Thank you for responding to my comments. In my opinion, the article is suitable for publication in its current form.

Response 1: Thank you very much for your favorable review and for deeming the article suitable for publication in its current form. Your positive feedback is greatly appreciated and confirms that our efforts to improve the manuscript have been successful. We are thrilled to have reached this stage.
